# The impact of cultural tourism experience on cultural identity: A case study of Mazu culture

**Mian Wang[1], Hee Seung Lee[2], Wen Qiang Chen** [1]*

**1** Department of Tourism Management, Putian University, Putian, China, **2** Department of Hotel Management, Honam University, Gwangju, Republic of Korea

* wenqiang1990@hotmail.com

## Abstract

Cultural heritage tourism is pivotal for fostering cultural identity, yet the psychological pathway from experience to identity formation in non-Western, folk-religious contexts remains unclear. This study examines this process within Mazu culture (a UNESCO intangible heritage), proposing a serial mediation model where 4E experiences (i.e., Entertainment, Educational, Escapist, Esthetic) foster emotional resonance, which in turn strengthens cultural identity, ultimately leading to place identity. Data from 583 tourists at Meizhou Island were analyzed using structural equation modeling and bootstrap tests. The findings reveal that all 4E dimensions significantly enhance emotional resonance, with the Escapist experience being the most potent. Crucially, the sequential mediation of emotional resonance and cultural identity was confirmed for 4E experiences, with the escapist experience exhibiting a complete mediation on place identity. Notably, Educational experience influenced cultural identity only indirectly via emotional resonance. As the first to empirically validate this complete "experience-emotion-identity" chain, this study refines the S-O-R paradigm by delineating its internal psychological sequence and highlights the paramount role of escapist immersion in folk-religious tourism, offering actionable insights for destination management and sustainable heritage preservation.

## Introduction

Cultural heritage tourism has become a pivotal pathway for constructing cultural identity and sustaining place attachment [1,2], driven by the growing global demand for authentic, immersive spiritual experiences [3]. Mazu culture—listed by UNESCO in 2009 as Intangible Cultural Heritage [4]—offers an ideal context for examining this process [5]. Syncretizing folk tradition with religious belief, Mazu worship spans 49 countries, 300 million adherents, and more than 10,000 temples [6]. This distinctive "folk-religious" context makes Mazu cultural tourism an ideal setting to explore how tourism experiences shape deep psychological outcomes like cultural and place identity—especially amid the commercialization of Chinese cultural heritage

**Data availability statement:** All raw data supporting the findings of this study are publicly available at the Open Science Framework (OSF) repository: https://osf.io/k6a7v/overview.

**Funding:** The authors acknowledge that the following projects provided partial financial support for the data collection (questionnaire survey) phase of this study: Research Project of Fujian Philosophy and Social Sciences Planning (FJ2025BF044); Youth Projects of the Social Science Fund of Jiangxi Province (25GL47); Putian Science and Technology Bureau Project (2023SZ3001PTXY12); Research Project of the Science and Technology Innovation Think Tank of the Fujian Provincial Association for Science and Technology (FJKX-2024XKB023); Startup Fund for Advanced Talents of Putian University (2021079). The funders had no role in study design, data analysis, decision to publish, or preparation of the manuscript.

**Competing interests:** The authors have declared that no competing interests exist.

tourism, which has left tourists' emotional responses and identity formation under-explored [7,8].

To unpack the "experience–emotion–identity" chain, we integrate a set of complementary frameworks. These theories synergistically underpin the proposed chained mediation model: The Stimulus–Organism–Response (S-O-R) framework [9,10] positions cultural tourism experience(4E) (Entertainment, Educational, Escapist, Esthetic) as "stimuli (S)", It reflects tourists' participatory and immersive engagement with cultural contexts [11–14], Emotional resonance and cultural identity as "organism (O)", A profound psychological response triggered by the alignment of tourism experiences with individuals' values and emotional needs. It bridges external stimuli and internal identity formation [15–17]. Cultural identity Individuals' sense of belonging [18] and value recognition toward a specific cultural system, forged through cognitive and emotional engagement with cultural elements [2,7,19–21]. And place identity as the final "response (R)". A core dimension of place attachment, reflecting the symbolic, emotional connection between individuals and a destination [18,22–26].

Additionally, Social Identity Theory [27] provides the foundational understanding of cultural identity as an individual's sense of belonging to a cultural group [28]. Resonance theory [29], explains why value-congruent experiences arouse profound affect [16,25,30], Fredrickson's Broaden-and-Build Theory of Positive Emotions links such affect to enduring cultural identities [7,31,32], while place attachment theory [26,33] clarifies the cognitive-affective linkage between cultural belonging and place connection (i.e., "cultural identity → place identity") [34,35].

Despite growing scholarly interest, five interrelated research gaps remain. (1) The complete sequential pathway from tourism experience through emotion to cultural and place identity lacks empirical validation. While the mediating role of emotion in place identity is noted [36], and the experience–emotion–cultural identity link within specific cultural activities is underexplored [37,38], the complete chained mediation model remains untested [39]. (2) The 4E framework's application is limited in culturally specific settings. Its unique dimensions and links to psychological outcomes in cultural tourism are poorly understood [12], and related place attachment theories are rarely validated in religious cultural contexts [40]. (3) The distinct role of emotional resonance as a mediator is overlooked. Studies often treat emotion generically, failing to explain how specific emotions like resonance translate experiences into identity outcomes [12,15,17]. (4) The linkage between cultural and place identity is fragmented. While cultural identity drives other outcomes [41], its connection to place identity and its role within an emotion–identity chain are not well established [42]. (5) Quantitative evidence from non-Western, folk-religious contexts is scarce. Research is geographically biased [37,43], with few studies examining emotional and identity mechanisms in China's cultural heritage tourism [8] or testing integrated frameworks in folk cultural tourism [44,45].

Using Mazu cultural tourism as the empirical setting, we test a chained mediation model: 4E experiences → emotional resonance → cultural identity → place identity. Specifically, this study addresses the following questions: How do the four dimensions of Mazu cultural tourism 4E experiences differentially influence tourists'

emotional resonance? Do emotional resonance and cultural identity sequentially mediate the effect of 4E experiences on place identity? How does testing this chained model advance the S-O-R paradigm by unpacking the "black box" of identity formation in cultural tourism? Therefore, this study aims to achieve the following three objectives: (1) To examine how the four dimensions of Mazu cultural tourism 4E experiences differentially influence tourists' emotional resonance. (2) To test the sequential mediating roles of emotional resonance and cultural identity in the relationship between 4E experiences and place identity. (3) To advance the S-O-R paradigm by unpacking the internal psychological sequence "black box" of identity formation in folk-religious cultural tourism.

This study makes several key contributions. Theoretically, it seeks to provide the first empirical validation of the complete "experience–emotion–dual identity" chain, thereby elaborating the internal processes of the S-O-R framework. It extends the application of the 4E model to the understudied context of folk-religious tourism, delineates the distinct mediating role of emotional resonance from generic affective responses. Furthermore, it clarifies the linkage between cultural identity and place identity, offering quantitative evidence from a non-Western, religious-heritage setting. Practically, the findings are expected to provide insights for designing Mazu cultural tourism experiences that foster emotional engagement and cultural inheritance.

## Literature review and hypotheses development

### Cultural tourism experience

Cultural tourism experience is fundamentally conceptualized through the lens of the Experience Economy Theory, which asserts that the core value of tourism lies in creating multidimensional, immersive experiences—encompassing entertainment, educational, escapist, and esthetic (4E)—rather than merely providing tangible products or basic services [14]. This theory provides a general typology for measuring tourist experiences through four distinct dimensions [12,46]. Within the specific domain of cultural tourism, particularly in folk-religious contexts such as Mazu culture, this framework offers a robust basis for delineating and measuring experience components [11,13]. This application also extends the 4E framework to non-Western cultural tourism settings, enriching its contextual adaptability [8]. Entertainment involves deriving pleasure from watching, listening, or participating in activities such as festivals and interactive programs [47,48]. Educational entails the active absorption of knowledge and cultural understanding [10]. Escapist experiences are defined by active immersion in an alternative environment, allowing tourists to disengage from daily routines and pressures, a mechanism noted for generating strong emotional attachment [49–51]. Lastly, Esthetic experience involves the passive immersion in and appreciation of a destination's sensory and artistic appeal, a dimension decisive to overall tourist perception [52,53]. While offering a universal typology, the empirical validation of the 4E framework derives predominantly from Western or secular settings [13,14]. Consequently, its transferability to value-laden, spiritually-oriented heritage contexts like Mazu culture—where tourism is deeply intertwined with belief and collective identity—remains less understood [8], and the scholarly understanding of its unique dimensions and linkages to psychological outcomes in such settings is underdeveloped [12].

However, The application of the generic 4E framework to folk-religious tourism contexts invites a nuanced consideration of its boundaries. First, while offering a universal typology, the framework's empirical validation derives predominantly from Western or secular settings [13,14]. Its transferability to value-laden, spiritually-oriented heritage contexts like Mazu culture, where tourism is deeply intertwined with belief and collective identity, requires further examination [8]. Second, the nature and salience of the framework's dimensions may be reconstituted in such an environment. For instance, within Mazu's devotional context, the 'Escapist' experience likely encompasses not merely recreational diversion but a deeper quest for spiritual solace and connection with the sacred [40]. Similarly, the 'Educational' dimension may function less as passive knowledge acquisition and more as an engagement with narratives that reinforce core cultural values essential to identity formation [7]. Third, treating the framework as a static checklist may overlook the dynamic, processual nature through which these experiences interact to co-construct meaning and identity. Consequently, this study employs the 4E framework as a foundational yet adaptable lens for empirical

investigation within this distinct milieu. It posits the framework's relevance but seeks to elucidate how its dimensions manifest and whether they exert differential psychological effects in folk-religious tourism, thereby addressing a noted conceptual and contextual gap [12].

Positioned as the foundational external Stimulus (S) within the overarching S-O-R paradigm [9], the 4E experiences are proposed to initiate the psychological process. This proposition is further underpinned by Resonance Theory [16,29,30], which elucidates that value-congruent experiences trigger profound affective connections. Empirically, entertainment experience generates excitement that fosters resonance by matching psychological needs for pleasure [12,47,54]. Educational experience, by facilitating knowledge acquisition, can induce strong positive emotions and transform cognitive engagement into affective resonance [10,41,55]. Escapist experience, characterized by deep immersion, is noted for generating strong emotional attachment by fulfilling needs for relief and novelty [49,51]. Lastly, esthetic experience automatically triggers emotional arousal via artistic cues, with sensory pleasure elevating positive affective states [13,51,52]. Therefore, we propose the following initial hypotheses within our model:

H1: Entertainment experience positively impacts Emotional Resonance.
H2: Educational experience positively impacts Emotional Resonance.
H3: Escapist experience positively impacts Emotional Resonance.
H4: Esthetic experience positively impacts Emotional Resonance.

## Emotional resonance

Emotional resonance represents a profound psychological state that is pivotal in translating external cultural tourism experiences into internal identity outcomes [16,17]. It is defined as a deep affective connection arising from the alignment between experiential stimuli and an individual's inner values and desires, which is distinguished from generic emotional arousal by its intensity and duration [15,54]. Resonance Theory provides the foundational mechanism, positing that such value congruence triggers potent emotional responses that can lead to significant psychological outcomes [29,30]. Despite its recognized importance in translating experiences into internal outcomes [16,17], a significant portion of existing research tends to treat tourist emotion generically. This overlooks the need to delineate the specific mediating role of profound, value-congruent affective states like resonance in the formation of enduring identities [12,15]. Within the S-O-R framework, this study conceptualizes emotional resonance as the primary internal Organismic state (O). It functions as the immediate affective bridge, translating the external Stimulus of 4E experiences into the subsequent cognitive and evaluative processes that underlie identity formation. This role is particularly salient in cultural tourism, where culture-related activities inherently evoke deep value congruence [3].

Emotional resonance is theorized to influence two critical downstream outcomes. First, according to Fredrickson's (2001) Broaden-and-Build Theory [31], the positive emotions inherent in resonance are posited to expand cognitive and social resources, thereby facilitating the construction of a stable cultural identity [7,56]. This link is supported by evidence that profound, value-congruent emotional experiences foster cultural belonging and identification [15,57]. This dynamic is particularly evident in Mazu cultural tourism, where rituals, heritage landscapes, and symbolic activities amplify resonance with tourists' cultural values [24,54]. Second, in line with Place Attachment Theory [26], the profound affective connection characteristic of resonance can transform a physical destination into a symbolically significant entity, directly contributing to a sense of place identity [17,23,40,58].

Therefore, as the pivotal affective Organismic state, emotional resonance is hypothesized to directly influence both the subsequent cognitive state and the final outcome in our model:

H5: Emotional resonance has a positive impact on Cultural Identity.
H6: Emotional resonance has a positive impact on Place Identity.

## Cultural identity

Cultural identity is a pivotal psychological outcome in cultural tourism [59]. Grounded in Social Identity Theory [27,60], it encompasses an individual's sense of belonging and emotional attachment to a cultural system [10,21]. In tourism, this identity is forged through cognitive and emotional engagement with heritage [2,20,61], fostering cultural confidence, pride, and advocacy [7,59]. In folk-religious contexts like Mazu culture, landscapes and rituals can strengthen this identity by awakening shared memories among homologous tourists [62]. Within the S-O-R framework of the present study, cultural identity is conceptualized as the key secondary Organismic state (O). It is sequentially positioned to capture the crystallization of affective resonance (the primary O) into a stable sense of belonging to the Mazu cultural system, subsequently driving attachment to the physical place.

Two sets of antecedents to this cultural identity are theorized. First, building on Social Identity Theory [21,27], the four dimensions of 4E cultural tourism experience are posited to directly shape this internal state. Entertainment experience serves as a stimulating source of positive affect, lowering psychological barriers and fostering openness to cultural connection [43,47]. Educational experience provides a cognitive foundation, facilitating the internalization of cultural values through knowledge acquisition [10,41]. Escapist experience, through deep immersion, aligns tourists' values with the cultural environment, thereby reducing psychological distance [16,49]. Esthetic experience cultivates admiration and pride through sensory-affective responses to cultural symbols [7,52]. Second, as established in the preceding section, the affective state of emotional resonance is theorized to be a critical direct precursor to cultural identity (H5).

As the central cognitive-evaluative Organismic state, cultural identity is further posited to be a direct driver of the final Response in the model. Theoretically, this link is anchored in Place Attachment Theory [26], which holds that place identity forms through the symbolic meanings individuals assign to a location, and Collective Memory Theory [56,63], which posits shared cultural memories as a bridge to physical places. When an individual identifies with a culture that is embodied in a specific place, this profound psychological attachment naturally extends to that destination, integrating it into the self-concept [34,35]. Empirical studies, including those in non-Western folk-religious settings like Mazu culture, support this logic, showing that stronger cultural identification deepens place belonging and attachment [22,43,62,64].

Therefore, we propose the following hypotheses concerning cultural identity:

H7: Entertainment experience positively impacts Cultural Identity.

H8: Educational experience positively impacts Cultural Identity.

H9: Escapist experience positively impacts Cultural Identity.

H10: Esthetic experience positively impacts Cultural Identity.

H11: Cultural identity has a positive impact on Place Identity.

## Place identity

Place identity represents the culmination of the psychological chain from cultural tourism experiences to a deep-seated emotional bond with a destination [65]. It is defined as the symbolic and affective dimension of place attachment, where a physical location becomes integrated into an individual's self-identity [26,66,67]. In cultural tourism, this bond extends beyond geography to encompass the socially constructed meanings, values, and cultural symbols embedded within a place [43,68]. Engagement with cultural heritage, such as Mazu temples and rituals, can thus transform a site of visit into a place of personal significance by powerfully evoking a sense of belonging [22,54]. However, the sequential psychological pathway culminating in place identity remains partially obscured. While its connection to tourism experiences is recognized [11], the integrated mechanisms involving specific emotional (e.g., resonance) and cultural (e.g., cultural identity) mediators within a chained model are underexplored [39]. This is particularly evident in non-Western, folk-religious contexts where place attachment theories require further validation [8,40]. Within the S-O-R framework, this study conceptualizes place identity as the ultimate Response (R). It is defined as the emotional attachment and symbolic connection

tourists feel towards Meizhou Island as the birthplace of Mazu culture. This conceptualization allows for an empirical test of a complete psychological pathway, wherein place identity is shaped both directly by experiential stimuli and indirectly through sequential internal states.

As the terminal outcome, place identity is theorized to be directly preceded by multiple pathways. First, drawing on Place Attachment Theory [26,33], the four dimensions of the cultural tourism experience are posited to be fundamental antecedents. Entertainment experience fosters place identity by creating positive and engaging interactions that enhance affective connection [11,66]. Educational experience provides a cognitive foundation, enabling tourists to assign greater symbolic meaning to the destination [69]. Escapist experience, through deep immersion, fosters a sense of psychological belonging central to place identity [70]. Esthetic experience cultivates a direct symbolic connection through sensory pleasure and admiration [52,66]. Second, as established previously, the internal Organismic states of emotional resonance (H6) and cultural identity (H11) are hypothesized to be direct drivers of this final Response.

Therefore, we propose the following final set of direct-effect hypotheses concerning place identity:

H12: Entertainment experience has a positive impact on Place Identity.

H13: Educational experience has a positive impact on Place Identity.

H14: Escapist experience has a positive impact on Place Identity.

H15: Esthetic experience has a positive impact on Place Identity.

## The serial mediation of emotional resonance and cultural identity

Integrating the theoretical foundations established in the preceding sections, this study proposes a complete sequential mediation model. Guided by the overarching S-O-R framework, the model posits that the four dimensions of Mazu cultural tourism experience (4E) indirectly influence Place Identity through the serial mediation of Emotional Resonance and Cultural Identity. This proposed pathway synthesizes distinct theoretical mechanisms into a coherent psychological process.

The sequence is theoretically underpinned as follows. First, drawing on Resonance Theory [16,29], value-congruent 4E experiences are posited to trigger profound Emotional Resonance. Second, informed by Fredrickson's (1998, 2001) Broaden-and-Build Theory [31,32], this positive emotional resonance is theorized to broaden cognitive and social resources, thereby facilitating the construction of an enduring Cultural Identity [7,56]. Third, in line with Place Attachment Theory [35], this strengthened cultural identity, reflecting a sense of belonging to the Mazu cultural system, becomes symbolically and emotionally anchored to its physical birthplace, Meizhou Island, culminating in Place Identity [22,43].

Therefore, we formally hypothesize the complete serial mediation pathway for each dimension of the cultural tourism experience:

H16: Entertainment experience positively influences Place Identity through the serial mediation of Emotional Resonance and Cultural Identity.

H17: Educational experience positively influences Place Identity through the serial mediation of Emotional Resonance and Cultural Identity.

H18: Escapist experience positively influences Place Identity through the serial mediation of Emotional Resonance and Cultural Identity.

H19: Esthetic experience positively influences Place Identity through the serial mediation of Emotional Resonance and Cultural Identity.

## Theoretical research model

Integrating the theoretical foundations and hypothesized relationships developed above, this study proposes a sequential mediation model to decipher the psychological process from cultural tourism experiences to identity formation. The overarching Stimulus-Organism-Response (S-O-R) framework [9] structures the entire model, positioning the multidimensional Mazu cultural tourism 4E experiences as the external Stimulus, emotional resonance and cultural identity as internal Organismic states, and place identity as the ultimate Response.

 

Within this S-O-R paradigm, specific theories illuminate the core mechanisms. Resonance Theory [16,29] explains why value-congruent 4E experiences trigger profound emotional resonance [15,30]. Subsequently, Fredrickson's (2001, 1998) Broaden-and-Build Theory [31,32] posits that such positive emotions broaden cognition and build enduring personal resources, thereby facilitating the construction of cultural identity [7]. Crucially, the very concept of cultural identity is grounded in Social Identity Theory [27], which asserts that individuals derive a part of their self-concept from their membership in social groups. In the tourism context, this translates to a sense of belonging and value recognition toward a cultural system [21], a construct that is forged through cognitive and emotional engagement with heritage. Finally, Place Attachment Theory [26,33] provides the logical foundation for the final link, asserting that an individual's identification with a culture (cultural identity) can become symbolically and emotionally anchored to the physical setting of that culture, thus forging a robust place identity [35,43].

Consequently, this model investigates the direct effects within the proposed chain and, most critically, tests the complete chained-mediation pathway: that 4E experiences indirectly influence place identity through the sequential mediation of emotional resonance and cultural identity. This integrated model, capturing the complete "experience → emotion → cultural identity → place identity" sequence, is depicted in Fig 1.

## Methods

### Data design

The study employed a structured questionnaire to measure the key constructs. All items were measured on a five-point Likert scale (1 = strongly disagree, 5 = strongly agree). The measurement scales were adapted from well-established instruments in prior tourism and psychology literature to ensure content validity and reliability within the specific context of

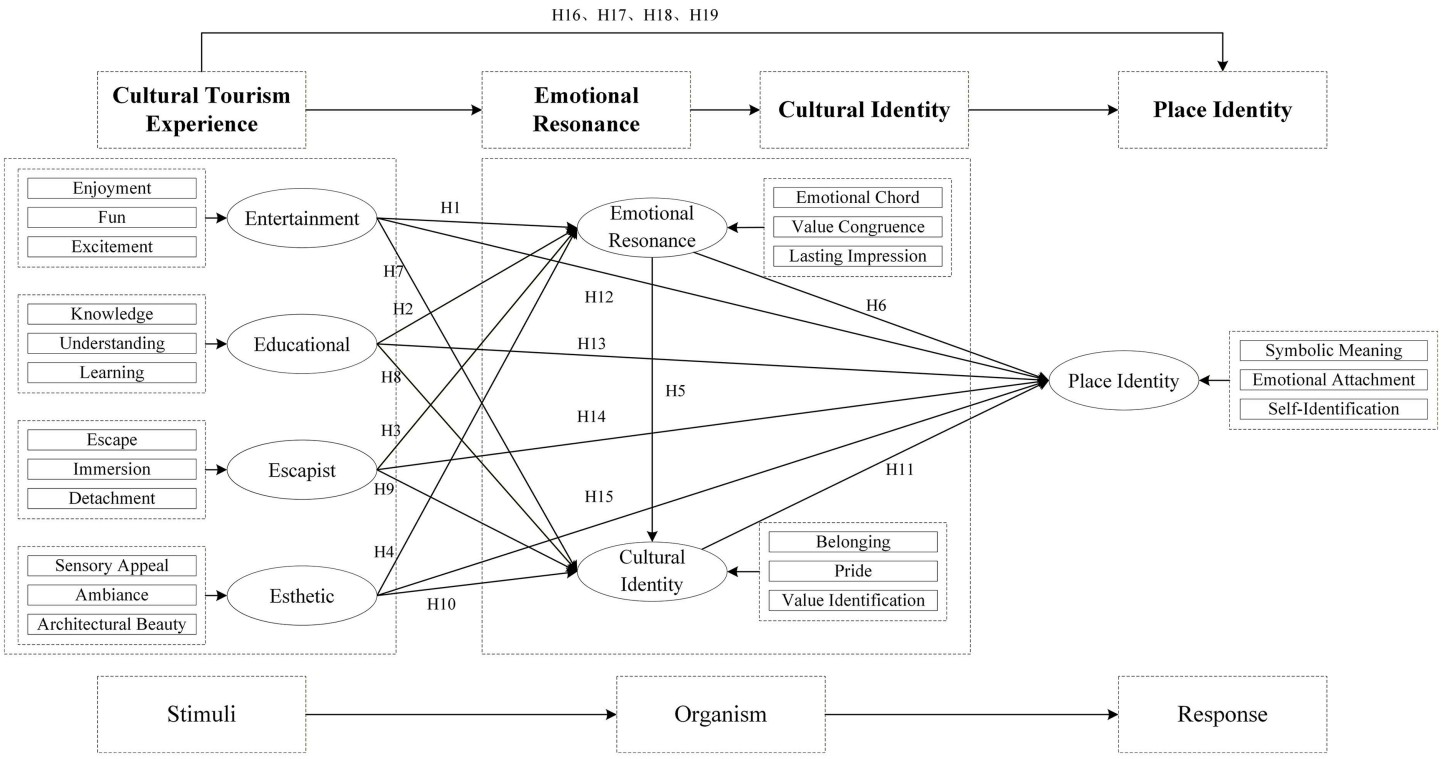

**Fig 1. Theoretical research model.**

Mazu cultural tourism. Specifically, the four dimensions of cultural tourism experience (4E) were adapted from Juliana et al. (2024) and Pine II and Gilmore (2016) [12,46]. Emotional Resonance items were sourced from Yan (2025) and Cheng et al. (2020) [15,17]. The Cultural Identity scale was adapted primarily from Yang et al. (2023), with reference to He and Ai (2021) for item conceptualization [7,59]. The Place Identity scale was adapted from Liu and Lin (2024) [66]. These scales were chosen for their strong psychometric properties and demonstrated applicability in tourism and cultural studies. Minor wording modifications were made to align all items explicitly with the Mazu cultural context of Meizhou Island (e.g., specifying "Mazu temples" to ensure contextual specificity). This design results in a parsimonious yet comprehensive measurement model, with each construct measured by three reflective indicators. Which provides a balance between model parsimony and adequate measurement for model identification, a common practice in SEM [71]. The complete wording of all measurement items is presented in Table 1. Detailed psychometric properties, including descriptive statistics, standardized factor loadings, and standard errors for all items, are provided in S1 Appendix.

## Data analysis

The collected data were analyzed in a sequence to test the hypothesized model. First, descriptive statistics and scale reliability were assessed using SPSS 26.0. Subsequently, covariance-based structural equation modeling (SEM) was employed using AMOS 24.0 to test the confirmatory, a priori theoretical model involving latent constructs, following the two-step approach of evaluating the measurement model prior to the structural model [71–73]. This method provides a comprehensive framework for simultaneously evaluating measurement and structural relationships and is a prevailing choice for such analytical purposes [74].

The measurement model was evaluated through confirmatory factor analysis (CFA) to establish reliability, convergent validity, and discriminant validity. The structural model was then examined to test the hypothesized paths (H1–H15).

**Table 1. Measurement items.**

| Construct | | Code | Item |
|---|---|---|---|
| Cultural Tourism Experience | Entertainment Experience | Ent1 | The Mazu cultural activities (e.g., festivals) were enjoyable. |
| | | Ent2 | I found the recreational activities at the Mazu sites to be fun. |
| | | Ent3 | Participating in these activities made me feel excited. |
| | Educational Experience | Edu1 | The visit helped me acquire knowledge about Mazu's history. |
| | | Edu2 | I gained a deeper understanding of Mazu beliefs and customs. |
| | | Edu3 | The experience enhanced my knowledge of Mazu culture. |
| | Escapist Experience | Esc1 | The Mazu cultural environment allowed me to escape from my daily routine. |
| | | Esc2 | I felt completely immersed in the spiritual atmosphere of Meizhou Island. |
| | | Esc3 | I experienced a sense of psychological detachment and spiritual transcendence here. |
| | Esthetic Experience | Est1 | I admired the sensory and artistic appeal of the Mazu temples and landscapes. |
| | | Est2 | The overall ambiance and esthetic of the Mazu sites was pleasing. |
| | | Est3 | I appreciated the architectural beauty and natural surroundings. |
| Emotional Resonance | | ER1 | My experience with Mazu culture struck a deep emotional chord with me. |
| | | ER2 | The values of Mazu culture resonated with my personal values. |
| | | ER3 | This cultural experience left a lasting emotional impression on me. |
| Cultural Identity | | CI1 | I feel a sense of belonging to the Mazu cultural community. |
| | | CI2 | I feel proud of the Mazu culture. |
| | | CI3 | I identify with the values of Mazu culture. |
| Place Identity | | PI1 | Meizhou Island, as the birthplace of Mazu, holds special symbolic meaning for me. |
| | | PI2 | I feel a strong emotional attachment to Meizhou Island. |
| | | PI3 | I feel that Meizhou Island is a part of my identity. |

 

Model fit was evaluated using multiple indices: χ²/df, SRMR, RMSEA, IFI, TLI, and CFI, with thresholds suggested in the literature (e.g., SRMR & RMSEA < 0.08; IFI/TLI/CFI > 0.90) [71,72]. To test the serial mediation hypotheses (H16–H19), the indirect effects were estimated using a bootstrap procedure with 2,000 resamples to obtain bias-corrected confidence intervals [75,76]. The final sample (N = 583) was deemed adequate for SEM, exceeding common recommendations for achieving stable parameter estimates [77].

## Data collection

**Study setting: Meizhou Island.** Meizhou Island, located in Putian City, Fujian Province, is widely recognized as the birthplace of Mazu culture [78]. It hosts the world's first Mazu temple, making it a spiritual epicenter for Mazu devotees worldwide. The "Mazu Belief and Customs" were inscribed on UNESCO's Representative List of the Intangible Cultural Heritage of Humanity in 2009 [5], highlighting its significance. Mazu belief has spread to 49 countries and regions, with over 10,000 affiliated temples and some 300 million adherents globally [6]. The island features rich Mazu-themed resources, including the Mazu Temple, Tianfei Hometown, Tianhou Palace, public squares, the Mazu Cultural Exhibition Hall, and traditional opera stages. These sites provide visitors with diverse and profound cultural experiences. Given its irreplaceable status and cultural tourism assets, Meizhou Island is an exemplary case for investigating the relationship between cultural tourism experiences and identity formation [79].

**Data collection and sample. Participants and Procedure**

A quantitative survey was administered to adult (aged 18 and above) domestic tourists on Meizhou Island from April to August 2025. Domestic tourists were targeted as they constitute the primary visitor group at this site of Chinese folk-religious heritage, ensuring contextual relevance for the study of Mazu cultural identity formation. This timeframe was selected due to its stable tourist flow, which helped minimize seasonal bias. A convenience sampling approach was used to obtain a diverse sample of visitors. Questionnaires were administered face-to-face by trained research assistants at high-traffic locations on the island, such as the exit of the main Mazu temple and visitor rest areas. The sample size was determined in accordance with the recommended subject-to-variable ratio for structural equation modeling [77].

**Ethical Considerations**

This study received ethical approval from the Ethics Committee of Putian University (Approval No.: Ethical Review (2025)020). The committee waived the requirement for written informed consent due to the anonymous and minimal-risk nature of the survey. Informed consent was obtained procedurally: a detailed cover page presented to all potential participants outlined the research purpose, procedures, data usage, voluntary nature, confidentiality measures, and the right to withdraw. Proceeding to complete and submit the questionnaire was considered implied consent. No personally identifiable information was collected.

**Sample Size and Validity**

Of the 620 questionnaires distributed, 608 were returned, yielding a 98% return rate. After the removal of 25 responses due to incompleteness or inconsistencies, 583 valid questionnaires were retained for analysis, rendering a 94% effective rate. The raw survey data are available in S2 File research data.

## Results

### Sample demographics

The demographic characteristics of the 583 valid participants are summarized in Table 2.

### Reliability and validity testing of scales

The reliability and validity of the measurement model were examined, employing SPSS 26.0 to perform exploratory factor analysis and AMOS 24.0 for confirmatory factor analysis (CFA), as detailed in Table 3. All scales demonstrated

**Table 2. Demographic profile of the sample (N = 583).**

| Demographic factor | Category | Frequency | Percentage (%) |
|---|---|---|---|
| Gender | Male | 234 | 40.1 |
| | Female | 349 | 59.9 |
| Education | Less than high school | 55 | 9.4 |
| | High school | 122 | 20.9 |
| | Bachelor's/Associate degree | 339 | 58.1 |
| | Master's degree or above | 67 | 11.5 |
| Monthly Income | RMB 3,000 or less | 214 | 36.7 |
| | RMB 3,001–6,000 | 198 | 34.0 |
| | RMB 6,001–9,000 | 115 | 19.7 |
| | RMB 9,001 or above | 56 | 9.6 |
| Age | 18-30 years old | 214 | 36.7 |
| | 31-40 years old | 100 | 17.2 |
| | 41-50 years old | 118 | 20.2 |
| | 51-60 years old | 113 | 19.4 |
| | 61 years old or above | 38 | 6.5 |
| Occupation | Student | 118 | 20.2 |
| | Employee of public institution | 99 | 17.0 |
| | Corporate employee | 157 | 26.9 |
| | Self-employed business owner | 116 | 19.9 |
| | Freelancer | 93 | 16.0 |

**Table 3. Reliability and validity test results of the scale (N = 583).**

| Construct | | Code | Mean | Exploratory Factor Analysis | | Confirmatory Factor Analysis | | | α |
|---|---|---|---|---|---|---|---|---|---|
| | | | | Loading | Variance Contribution (%) | Loading | CR | AVE | |
| Cultural Tourism Experience | Entertainment Experience | Ent1 | 3.79 | 0.871 | 18.06 | 0.749 | 0.807 | 0.582 | 0.806 |
| | | Ent2 | 3.64 | 0.795 | | 0.799 | | | |
| | | Ent3 | 3.68 | 0.768 | | 0.739 | | | |
| | Educational Experience | Edu1 | 3.71 | 0.845 | 18.201 | 0.780 | 0.824 | 0.609 | 0.823 |
| | | Edu2 | 3.75 | 0.785 | | 0.747 | | | |
| | | Edu3 | 3.77 | 0.783 | | 0.813 | | | |
| | Escapist Experience | Esc1 | 3.59 | 0.846 | 18.765 | 0.833 | 0.828 | 0.617 | 0.826 |
| | | Esc2 | 3.65 | 0.815 | | 0.748 | | | |
| | | Esc3 | 3.62 | 0.828 | | 0.773 | | | |
| | Esthetic Experience | Est1 | 3.57 | 0.815 | 17.335 | 0.766 | 0.757 | 0.511 | 0.753 |
| | | Est2 | 3.59 | 0.789 | | 0.710 | | | |
| | | Est3 | 3.60 | 0.737 | | 0.664 | | | |
| Emotional Resonance | | ER1 | 3.61 | 0.792 | 67.886 | 0.634 | 0.765 | 0.522 | 0.763 |
| | | ER2 | 3.51 | 0.833 | | 0.734 | | | |
| | | ER3 | 3.52 | 0.846 | | 0.791 | | | |
| Cultural Identity | | CI1 | 3.62 | 0.858 | 74.798 | 0.767 | 0.832 | 0.622 | 0.831 |
| | | CI2 | 3.65 | 0.875 | | 0.815 | | | |
| | | CI3 | 3.64 | 0.862 | | 0.784 | | | |
| Place Identity | | PI1 | 4.34 | 0.828 | 69.195 | 0.708 | 0.777 | 0.538 | 0.777 |
| | | PI2 | 4.36 | 0.828 | | 0.749 | | | |
| | | PI3 | 4.31 | 0.840 | | 0.742 | | | |

satisfactory internal reliability. The computed Cronbach's alpha coefficients returned values between 0.753 and 0.831, each surpassing the 0.70 benchmark [80]. An exploratory factor analysis revealed that all item loadings exceeded 0.70. For the cultural tourism experience, four factors were extracted: "Entertainment Experience", "Educational Experience" "Escapist Experience" and "Esthetic Experience" The KMO value was 0.839 (p = 0.000), For the remaining constructs (KMO values: Emotional Resonance = 0.688, Cultural Identity = 0.723, Place Identity = 0.702), and the total variance explained for the core constructs ranged from 67.89% to 74.80%, supporting their unidimensionality [81]. The CFA results indicated excellent model fit ($\chi^2$/df = 1.720, SRMR = 0.032, RMSEA = 0.035, IFI = 0.977, TLI = 0.971, CFI = 0.977) All fit indices meet or surpass the recommended thresholds for a good fit (e.g., SRMR & RMSEA < 0.08; IFI/TLI/CFI > 0.90) [72]. Furthermore, all constructs demonstrated satisfactory convergent validity, with composite reliability (CR) values ranging from 0.757 to 0.832 and average variance extracted (AVE) estimates between 0.511 and 0.622, each exceeding their respective benchmarks of 0.70 and 0.50 [82]. Finally, potential common method bias (CMB) was assessed using two complementary approaches. Harman's single-factor test revealed that the single extracted factor accounted for 36.1% of the total variance (<50%). A marker variable test was also conducted: comparison between the baseline model ($\chi^2$ = 355.821, df = 224) and the model containing the marker variable ($\chi^2$ = 355.734, df = 223) showed no significant fit improvement ($\Delta\chi^2$(1) = 0.087, p > 0.05). These results collectively suggest that common method bias is not a major concern for this study [83]. As summarized in Table 3, all constructs demonstrated satisfactory reliability and validity. Detailed item-level statistics, including means, standard deviations, standardized factor loadings, and standard errors for all measurement items, are provided in S1 Appendix.

Discriminant validity was assessed using two established criteria. First, as shown in Table 4, the square root of each construct's AVE (diagonal) exceeded its correlations with other constructs (off-diagonal), satisfying the Fornell-Larcker criterion [84]. Second, we computed the Heterotrait-Monotrait (HTMT) ratio. As presented in Table 5, all HTMT values ranged from 0.401 to 0.670, well below the conservative threshold of 0.85. Furthermore, As presented in Table 6 the 95% bias-corrected bootstrap confidence intervals for these ratios did not include 1 [85]. In summary, these findings confirm the adequacy of the measurement model, which exhibits robust psychometric properties, making it suitable for subsequent analysis.

## Structural equation modeling testing results

Fig 2 illustrates the proposed serial mediation model with standardized path coefficients constructed using AMOS 24.0. The hypothesized model (Model 1) demonstrated an excellent fit to the data: $\chi^2$/df = 1.720, SRMR = 0.032, RMSEA = 0.035, IFI = 0.977, TLI = 0.971, CFI = 0.977. All indices meet or surpass the recommended thresholds for a good model fit (e.g., $\chi^2$/

**Table 4. Correlation coefficients and the square root of average variance extracted (AVE) (N = 583).**

| Variable | Ent | Edu | Esc | Est | ER | CI | PI |
|---|---|---|---|---|---|---|---|
| Ent | **0.763** | | | | | | |
| Edu | 0.517*** | **0.780** | | | | | |
| Esc | 0.461*** | 0.455*** | **0.785** | | | | |
| Est | 0.391*** | 0.607*** | 0.396*** | **0.715** | | | |
| ER | 0.495*** | 0.541*** | 0.521*** | 0.497*** | **0.722** | | |
| CI | 0.483*** | 0.486*** | 0.482*** | 0.482*** | 0.602*** | **0.789** | |
| PI | 0.631*** | 0.619*** | 0.513*** | 0.566*** | 0.629*** | 0.67*** | **0.733** |

a Bold values on the matrix diagonal represent the square root of AVE. Below-diagonal figures indicate variable correlation coefficients, with ***denoting p < 0.001 significance.

b Ent = Entertainment Experience; Edu = Educational Experience; Esc = Escapist Experience; Est = Esthetic Experience; ER = Emotional Resonance; CI = Cultural Identity; PI = Place Identity.

**Table 5. Heterotrait-Monotrait ratio (HTMT).**

| Variable | Ent | Edu | Esc | Est | ER | CI | PI |
|---|---|---|---|---|---|---|---|
| Ent | | | | | | | |
| Edu | 0.512 | | | | | | |
| Esc | 0.461 | 0.456 | | | | | |
| Est | 0.401 | 0.613 | 0.410 | | | | |
| ER | 0.496 | 0.545 | 0.522 | 0.506 | | | |
| CI | 0.487 | 0.486 | 0.490 | 0.491 | 0.592 | | |
| PI | 0.627 | 0.613 | 0.519 | 0.564 | 0.633 | 0.670 | |

[a]Ent = Entertainment Experience; Edu = Educational Experience; Esc = Escapist Experience; Est = Esthetic Experience; ER = Emotional Resonance; CI = Cultural Identity; PI = Place Identity.

**Table 6. HTMT$_{inference}$.**

| | Original sample (O) | Sample mean (M) | 5.00% | 95.00% | Bias | 5.00% | 95.00% |
|---|---|---|---|---|---|---|---|
| Ent<->ER | 0.496 | 0.495 | 0.417 | 0.57 | 0 | 0.415 | 0.568 |
| Edu<->ER | 0.545 | 0.544 | 0.471 | 0.614 | 0 | 0.469 | 0.613 |
| Esc<->ER | 0.522 | 0.522 | 0.445 | 0.599 | 0.001 | 0.442 | 0.597 |
| Est<->ER | 0.506 | 0.506 | 0.421 | 0.587 | −0.001 | 0.418 | 0.584 |
| ER<->CI | 0.592 | 0.591 | 0.509 | 0.668 | −0.001 | 0.507 | 0.666 |
| ER<->PI | 0.633 | 0.632 | 0.562 | 0.698 | −0.001 | 0.561 | 0.697 |
| Ent<->CI | 0.487 | 0.488 | 0.411 | 0.562 | 0.001 | 0.407 | 0.559 |
| Edu<->CI | 0.486 | 0.486 | 0.407 | 0.564 | 0 | 0.405 | 0.563 |
| Esc<->CI | 0.49 | 0.491 | 0.419 | 0.559 | 0 | 0.418 | 0.557 |
| Est<->CI | 0.491 | 0.491 | 0.411 | 0.567 | 0 | 0.409 | 0.565 |
| CI <->PI | 0.67 | 0.671 | 0.605 | 0.736 | 0.001 | 0.599 | 0.731 |
| Ent<->PI | 0.627 | 0.627 | 0.565 | 0.688 | 0 | 0.564 | 0.687 |
| Edu<->PI | 0.613 | 0.613 | 0.545 | 0.679 | 0 | 0.543 | 0.678 |
| Esc<->PI | 0.519 | 0.519 | 0.444 | 0.592 | 0 | 0.441 | 0.59 |
| Est<->PI | 0.564 | 0.565 | 0.496 | 0.632 | 0.001 | 0.492 | 0.628 |
| Ent<->Edu | 0.512 | 0.512 | 0.437 | 0.585 | 0 | 0.436 | 0.584 |
| Ent<->Esc | 0.461 | 0.461 | 0.388 | 0.533 | 0 | 0.387 | 0.532 |
| Ent<->Est | 0.401 | 0.401 | 0.32 | 0.482 | 0 | 0.32 | 0.481 |
| Edu<->Esc | 0.456 | 0.456 | 0.377 | 0.534 | 0 | 0.378 | 0.534 |
| Edu<->Est | 0.613 | 0.612 | 0.542 | 0.68 | −0.001 | 0.545 | 0.682 |
| Esc<->Est | 0.41 | 0.41 | 0.327 | 0.49 | 0 | 0.325 | 0.488 |

[a]Ent = Entertainment Experience; Edu = Educational Experience; Esc = Escapist Experience; Est = Esthetic Experience; ER = Emotional Resonance; CI = Cultural Identity; PI = Place Identity.

df < 3, SRMR & RMSEA < 0.08, IFI/TLI/CFI > 0.90), collectively supporting the robustness of the model [72,86]. To further justify the proposed sequential mediation pathway, we compared Model 1 against two theoretically derived, nested alternative models: a model with emotional resonance as the sole mediator (Model 2: 4E→ER→PI) and a model with cultural identity as the sole mediator (Model 3: 4E→CI→PI). The fit indices for all models are presented in Table 7. The model comparisons yield three key insights. First, the hypothesized sequential model (Model 1) demonstrated the highest explanatory power for place identity ($R^2$ = .651). Second, it provided a statistically equivalent fit to the more parsimonious

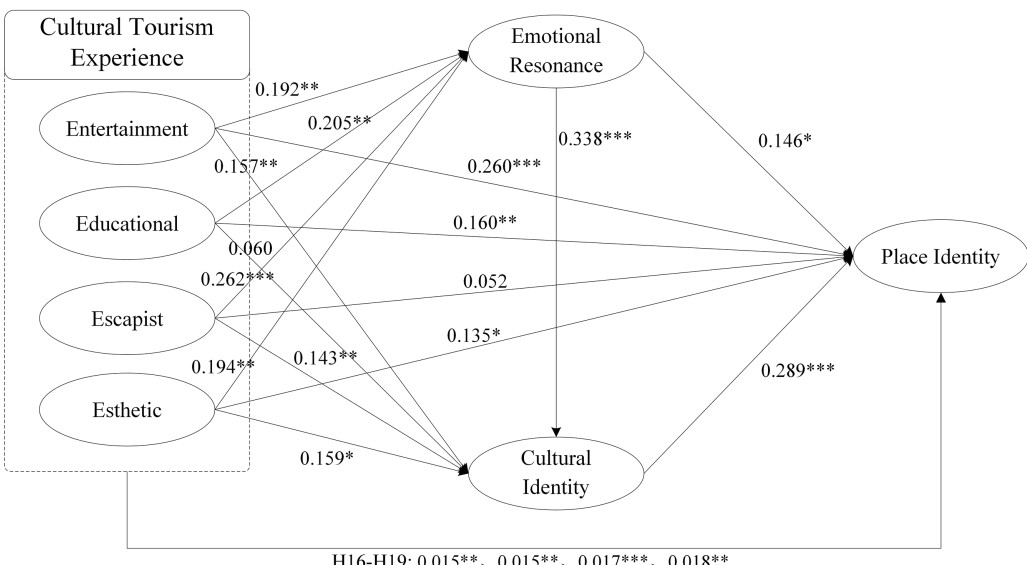

**Fig 2. Examination results of structural equation model.**

**Table 7. Model fit comparison.**

| Model | Path | χ² | df | χ²/df | SRMR | RMSEA | IFI | TLI | CFI | R² |
|---|---|---|---|---|---|---|---|---|---|---|
| Model 1 | 4E→ER→CI→PI | 288.888 | 168 | 1.720 | 0.032 | 0.035 | 0.977 | 0.971 | 0.977 | 0.651 |
| Model 2 | 4E→ER→PI | 230.657 | 120 | 1.922 | 0.033 | 0.040 | 0.974 | 0.967 | 0.974 | 0.606 |
| Model 3 | 4E→CI→PI | 213.686 | 120 | 1.781 | 0.033 | 0.037 | 0.979 | 0.973 | 0.979 | 0.641 |

[a]4E = Cultural Tourism Experience; ER = Emotional Resonance; CI = Cultural Identity; PI = Place Identity.

Model 2 (Δχ²(48) = 58.231, p > .05), indicating no penalty in fit for including the additional mediator. Crucially, however, Model 1 provided a significantly better fit than Model 3 (Δχ²(48) = 75.202, p < .05), underscoring the indispensable role of emotional resonance as the initial stage in the mediation chain.

Collectively, these results support the robustness of the hypothesized model. It is retained as the preferred representation of the data because it provides the most complete theoretical account of the psychological sequence—from affective experience to identity internalization—without compromising model parsimony, while also offering the greatest predictive validity.

As summarized in Table 8, the path analysis revealed that all four dimensions of the Mazu cultural tourism experience significantly enhanced emotional resonance: Entertainment (β = 0.192, t = 3.287, p = 0.001), Educational (β = 0.205, t = 2.982, p = 0.003), Escapist (β = 0.262, t = 4.666, p < 0.001), and Esthetic (β = 0.194, t = 2.984, p = 0.003), thus supporting H1–H4. In predicting cultural identity, emotional resonance showed a significant positive effect (β = 0.338, t = 5.116, p < 0.001), supporting H5. Regarding the direct effects of experiences, Entertainment (β = 0.157, t = 2.798, p = 0.005), Escapist (β = 0.143, t = 2.640, p = 0.008), and Esthetic (β = 0.159, t = 2.558, p = 0.011) experiences were significant, providing support for H7, H9, and H10 respectively. However, the path from Educational experience to cultural identity was not significant (β = 0.060, t = 0.916, p = 0.360); therefore, H8 was not supported.

**Table 8. SEM testing results.**

| Outcome Variable | Predictive Variable | R² | β | SEs | t | P | LLCI | ULCI |
|---|---|---|---|---|---|---|---|---|
| Equation 1 | | | | | | | | |
| ER | Ent | 0.439 | 0.192 | 0.054 | 3.287 | 0.001 | 0.064 | 0.305 |
| ER | Edu | | 0.205 | 0.060 | 2.982 | 0.003 | 0.069 | 0.348 |
| ER | Esc | | 0.262 | 0.044 | 4.666 | *** | 0.147 | 0.374 |
| ER | Est | | 0.194 | 0.072 | 2.984 | 0.003 | 0.062 | 0.344 |
| Equation 2 | | | | | | | | |
| CI | Ent | 0.454 | 0.157 | 0.061 | 2.798 | 0.005 | 0.044 | 0.278 |
| CI | Edu | | 0.060 | 0.067 | 0.916 | 0.360 | −0.086 | 0.202 |
| CI | Esc | | 0.143 | 0.050 | 2.640 | 0.008 | 0.033 | 0.249 |
| CI | Est | | 0.159 | 0.082 | 2.558 | 0.011 | 0.028 | 0.296 |
| CI | ER | | 0.338 | 0.078 | 5.116 | *** | 0.155 | 0.512 |
| Equation 3 | | | | | | | | |
| PI | Ent | 0.651 | 0.260 | 0.042 | 4.788 | *** | 0.150 | 0.371 |
| PI | Edu | | 0.160 | 0.044 | 2.622 | 0.009 | 0.020 | 0.290 |
| PI | Esc | | 0.052 | 0.033 | 1.027 | 0.305 | −0.058 | 0.167 |
| PI | Est | | 0.135 | 0.055 | 2.295 | 0.022 | 0.013 | 0.257 |
| PI | ER | | 0.146 | 0.053 | 2.320 | 0.020 | −0.004 | 0.289 |
| PI | CI | | 0.289 | 0.041 | 4.948 | *** | 0.169 | 0.420 |

a Note: ***$p < 0.001$, **$p < 0.01$, *$p < 0.05$;

b Ent = Entertainment Experience; Edu = Educational Experience; Esc = Escapist Experience; Est = Esthetic Experience; ER = Emotional Resonance; CI = Cultural Identity; PI = Place Identity.

For place identity, significant direct predictors included: Entertainment experience (β = 0.260, t = 4.788, p < 0.001), supporting H12; Educational experience (β = 0.160, t = 2.622, p = 0.009), supporting H13; Esthetic experience (β = 0.135, t = 2.295, p = 0.022), supporting H15; emotional resonance (β = 0.146, t = 2.320, p = 0.020), supporting H6; and cultural identity (β = 0.289, t = 4.948, p < 0.001), supporting H11. The direct path from Escapist experience to place identity was not significant (β = 0.052, t = 1.027, p = 0.305), thus H14 was not supported.

## Serial mediation test results

To test the hypothesized sequential mediation model, a bootstrap analysis with 2000 resamples was conducted using AMOS 24.0, with significance determined by 95% bias-corrected confidence intervals (BC CI) not containing zero [75,76]. The establishment of serial mediation requires a statistically significant specific indirect effect for the path (X→Mediator1→Mediator2→Y). A complete serial mediation is evidenced when the direct effect (X→Y) becomes non-significant after including the mediators. In contrast, partial serial mediation occurs if the direct effect remains significant [71].

As summarized in Table 9, the serial mediation through emotional resonance and cultural identity was supported for all four experience dimensions. The total indirect effects on place identity were significant for entertainment (effect = 0.071, 95% BC CI [0.034, 0.122]), educational (effect = 0.049, 95% BC CI [0.010, 0.096]), escapist (effect = 0.069, 95% BC CI [0.039, 0.112]), and esthetic experiences (effect = 0.087, 95% BC CI [0.039, 0.159]). Crucially, the specific indirect effects for the path Experience→Emotional Resonance→Cultural Identity→Place Identity were all statistically significant: entertainment (effect = 0.015, 95% BC CI [0.005, 0.035]), educational (effect = 0.015, 95% BC CI [0.004, 0.037]), escapist (effect = 0.017, 95% BC CI [0.007, 0.037]), and esthetic (effect = 0.018, 95% BC CI [0.005, 0.043]). Notably, for the escapist experience, the direct effect on place identity was non-significant (β = 0.052, p = 0.305, from Table 5; and direct effect 95% BC CI [−0.040, 0.108]), suggesting a complete mediation mechanism. For the other three experience dimensions,

**Table 9. Serial mediation test results.**

| Effect types | Effect | Boot SE | Boot LLCI | Boot ULCI | p | Ratio of indirect to total effect | Ratio of indirect to direct effect |
|---|---|---|---|---|---|---|---|
| Total effect | 0.273 | 0.048 | 0.185 | 0.373 | 0.001 | – | – |
| Direct effect | 0.201 | 0.048 | 0.113 | 0.299 | 0.001 | – | – |
| Total indirect effect | 0.071 | 0.022 | 0.034 | 0.122 | 0.001 | 26.01% | 35.32% |
| Ent→ER→PI | 0.022 | 0.014 | 0.001 | 0.058 | 0.039 | 8.06% | 10.95% |
| Ent→CI→PI | 0.035 | 0.017 | 0.009 | 0.076 | 0.006 | 12.82% | 17.41% |
| Ent→ER→CI→PI | 0.015 | 0.007 | 0.005 | 0.035 | 0.002 | 5.49% | 7.46% |
| Total effect | 0.166 | 0.049 | 0.068 | 0.263 | 0.002 | – | – |
| Direct effect | 0.117 | 0.048 | 0.014 | 0.207 | 0.027 | – | – |
| Total indirect effect | 0.049 | 0.022 | 0.01 | 0.096 | 0.012 | 29.52% | 41.88% |
| Edu→ER→PI | 0.022 | 0.014 | 0.001 | 0.061 | 0.031 | 13.25% | 18.80% |
| Edu→CI→PI | 0.013 | 0.016 | −0.018 | 0.046 | 0.381 | 7.83% | 11.11% |
| Edu→ER→CI→PI | 0.015 | 0.008 | 0.004 | 0.037 | 0.003 | 9.04% | 12.82% |
| Total effect | 0.103 | 0.039 | 0.023 | 0.179 | 0.014 | – | – |
| Direct effect | 0.034 | 0.039 | −0.04 | 0.108 | 0.358 | – | – |
| Total indirect effect | 0.069 | 0.019 | 0.039 | 0.112 | 0.001 | 66.99% | 202.94% |
| Esc→ER→PI | 0.025 | 0.014 | 0.001 | 0.058 | 0.044 | 24.27% | 73.53% |
| Esc→CI→PI | 0.027 | 0.014 | 0.007 | 0.06 | 0.009 | 26.21% | 79.41% |
| Esc→ER→CI→PI | 0.017 | 0.007 | 0.007 | 0.037 | 0 | 16.50% | 50.00% |
| Total effect | 0.212 | 0.06 | 0.104 | 0.336 | 0.001 | – | – |
| Direct effect | 0.125 | 0.06 | 0.013 | 0.251 | 0.028 | – | – |
| Total indirect effect | 0.087 | 0.03 | 0.039 | 0.159 | 0.001 | 41.04% | 69.60% |
| Est→ER→PI | 0.026 | 0.018 | 0.002 | 0.075 | 0.033 | 12.26% | 20.80% |
| Est→CI→PI | 0.043 | 0.023 | 0.009 | 0.105 | 0.016 | 20.28% | 34.40% |
| Est→ER→CI→PI | 0.018 | 0.009 | 0.005 | 0.043 | 0.001 | 8.49% | 14.40% |

[a]Ent = Entertainment Experience; Edu = Educational Experience; Esc = Escapist Experience; Est = Esthetic Experience; ER = Emotional Resonance; CI = Cultural Identity; PI = Place Identity.

the direct effects remained significant, indicating partial serial mediation. These findings robustly confirm that emotional resonance and cultural identity sequentially transmit the influence of 4E experiences to place identity, thereby fully supporting H16–H19.

## Discussion

This study provides empirical support for a sequential mediation model in which 4E tourism experiences foster emotional resonance, which in turn strengthens cultural identity, ultimately leading to place identity within the folk-religious context of Mazu culture. By delineating this specific psychological pathway, the findings help to unpack the "black box" linking multidimensional tourism experiences to the formation of dual identities [39,42]. The results indicate a mechanism that may be more nuanced than the established view of emotion as a generic mediator [36], highlighting emotional resonance and cultural identity as sequential and critical mediators. This perspective extends the Stimulus-Organism-Response (S-O-R) framework [9] by proposing that the "organism" component may be better conceptualized as a dynamic, two-stage process — involving initial affective alignment (resonance) followed by cognitive-identity internalization [17] — particularly attuned to the value-rich context of folk-religious tourism. This sequential mediation mechanism aligns with the preliminary findings on the experience-emotion-identity segmented path in cultural tourism [87,88].

## Interpreting the centrality of escapist and affective pathways

A pivotal finding is the superior potency of the escapist experience in evoking emotional resonance (β = 0.262, p < 0.001) and its role as a complete mediator for place identity (H18 supported; direct effect n.s.). This underscores that in Mazu cultural tourism, deep immersion and psychological detachment are not merely enjoyable but foundational for meaning-making [51]. As Angeloni (2023) argued, such immersive experiences generate strong emotional attachment through the fulfillment of needs for relief and novelty [49]. This process facilitates the creation of a perceived "sacred space" [40] on Meizhou Island, where the tourist's search for spiritual solace and connection aligns with the cultural system's core values (e.g., Mazu's benevolence and protection), thereby triggering profound resonance [16,29]. This suggests that for deeply symbolic cultural experiences, the path to place attachment is fully mediated by this internal affective-cognitive transformation, a nuance not fully captured in standard S-O-R applications within tourism studies [11]. The dominant role of escapist experience in evoking emotional resonance is also supported by Juliana et al. (2024) [12], and its complete mediation effect on place identity further supplements the findings of Allan (2016) and Liu and Lin (2024) in cultural tourism contexts [11,66].

## The indirect role of educational experience: A contextual nuance

The non-significant direct path from educational experience to cultural identity (H8 not supported) presents a critical contextual nuance specific to this folk-religious setting. While knowledge acquisition is often a primary identity driver in other heritage contexts [41], our findings point to a different mechanism in Mazu cultural tourism. Here, passive learning about history or customs appears insufficient to directly forge a deep sense of belonging. Instead, its primary role is to supply the cognitive schemata (narratives and values) essential for deeper engagement. Its significant effect on emotional resonance (H2 supported) suggests that identity formation in this value-laden context is gated by affective evaluation: knowledge must resonate with tourists' personal values (e.g., safety, family) to spark the emotional connection that, in turn, builds identity [31]. This interpretation is consistent with Zhang and Liu (2025), who found that educational experiences in cultural tourism require emotional internalization to influence identity outcomes [10]. This delineates an important boundary condition, extending social identity theory by highlighting a context where affective alignment precedes cognitive internalization [27]. This indirect effect of educational experience on cultural identity contrasts with the direct effect reported by Urošević (2012) in general cultural tourism contexts [43]. The divergence suggests that in folk-religious settings, identity formation follows an 'affective priority' pathway, wherein knowledge must first achieve emotional resonance to translate into cultural belonging—a nuance that extends social identity theory to value-laden heritage contexts.

**Emotional resonance as the specific conduit.** The robust effect of emotional resonance on both cultural identity (H5 supported) and place identity (H6 supported) confirms its distinct role as the crucial bridge in the S-O-R sequence. Unlike transient positive affect, resonance—defined by value alignment and lasting impression [15]—appears to provide the sustained emotional energy necessary for building enduring psychological assets like identity. This finding challenges the treatment of emotion as a generic mediator and positions emotional resonance as a specific and pivotal mechanism for translating experiential stimuli into deep-seated identity outcomes within cultural tourism. The positive effect of emotional resonance on cultural identity and place identity is strongly supported by Qu et al. (2025) [24], Yang et al. (2023) and Hosany et al. (2016), verifying its universal mediating value in cross-cultural tourism contexts [7,23]. Furthermore, Wang et al. (2023) demonstrated that emotional resonance directly enhances place identity, a finding corroborated by our results [89].

The significant direct effects of entertainment and aesthetic experiences on cultural identity and place identity are consistent with the empirical results of Urošević (2012) [43], Juliana et al. (2024) and Allan (2016), which further verify the universal positive role of these two experience dimensions in shaping tourists' dual identity in cultural tourism [11,12]. In addition, the positive driving effect of cultural identity on place identity confirmed in this study is in line with Casais and

Poço (2023) [64], highlighting that cultural identity is the core cognitive bridge connecting cultural experience and place attachment across different tourism contexts. This finding is further supported by Cao et al. (2021) [22], who found that tourists' identification with heritage culture enhances their emotional bond with the physical site.

## Theoretical Implications

The findings of this study carry significant implications for several theoretical domains in tourism and heritage studies.

**Elaborating the S-O-R framework: A two-stage organismic process.** The validated sequential model moves beyond treating the S-O-R's "organism" (O) as a monolithic internal state. Instead, it proposes that in value-laden, folk-religious tourism, the O component is best understood as a dynamic, two-stage sequence (Affective Alignment→Identity Internalization). This specification addresses a key ambiguity in applying the S-O-R paradigm to complex cultural experiences and answers calls for greater precision in modeling internal psychological processes in tourism [9]. By delineating emotional resonance as the immediate affective bridge and cultural identity as the subsequent cognitive crystallization, the study refines the theoretical "black box" of the O state.

**Contextualizing and hierarchizing the 4E experience economy model.** This research extends the application of the 4E framework [14] to the understudied domain of folk-religious tourism, revealing that its dimensions are not equally potent. The findings demonstrate a hierarchy of influence, with the Escapist dimension emerging as the most potent catalyst for emotional resonance and serving as a complete mediator. This challenges the implicit assumption of dimensional equivalence in the model and underscores the need to consider the differential psychological efficacy of experience types based on the specific tourism context. The indirect role of the Educational dimension further suggests that the model's application must account for context-specific pathways to outcome variables like identity.

**Establishing an affective gate in cultural identity formation.** The non-significant direct effect of Educational experience on Cultural identity, coupled with its strong indirect effect through Emotional Resonance, establishes a crucial boundary condition for Social Identity Theory in tourism [27]. It reveals that in spiritually-oriented, symbolic heritage contexts, cognitive engagement (through education) is a necessary but insufficient condition for building cultural identity. Identity formation is effectively "gated" by an affective evaluation process, where knowledge must achieve value congruence to spark the emotional resonance that fuels identification. This "affective primacy" pathway enriches our understanding of how cultural identity is constructed in non-Western, value-driven settings.

**A refined sequential S-O-R model.** Synthesizing these findings, we propose a theoretical model for identity formation in folk-religious cultural tourism. This model elaborates the S-O-R paradigm by specifying its internal sequence: the "Organism" is reconceptualized as a two-stage sequential process where external Stimuli (4E experiences) first trigger Emotional Resonance (Stage 1: Affective Alignment), which then facilitates the construction of Cultural Identity (Stage 2: Identity Internalization), ultimately leading to the Response of Place Identity. Furthermore, it contextualizes the 4E Framework: The efficacy of each dimension is contingent on its capacity to generate value-congruent resonance. Escapist experiences emerge as paramount; Educational experiences require emotional activation; Entertainment and Esthetic experiences serve as potent entry points but rely on the subsequent chain to forge lasting place bonds. Therefore, we advance the core proposition: In folk-religious cultural tourism, the path from experiential stimuli to place identity is most accurately characterized as a sequential mediation process, wherein multidimensional experiences foster emotional resonance, which in turn is essential for strengthening cultural identity, thereby culminating in place identity. This proposition contributes to addressing inconsistencies in prior literature by specifying the "how" and "in what order" of the identity formation process in this distinct context.

## Practical implications

The findings translate into actionable strategies for destination Management Organizations (DMOs) and site managers to enhance tourist identity formation and advance sustainable heritage preservation in Mazu cultural tourism.

**Prioritizing escapist experiences to cultivate sacred immersion.** Given the escapist experiences' complete mediation effect and its superior potency in evoking emotional resonance, DMOs should prioritize creating environments that foster psychological detachment and deep spiritual engagement. This can be achieved by curating guided "spiritual retreats" on Meizhou Island that facilitate participation in authentic rituals in contemplative settings, or employing immersive technologies (e.g., VR) to reconstruct pivotal historical narratives of Mazu, thereby enabling tourists to symbolically immerse themselves in the culture. These interventions align with the principle of prioritizing "emotional immersion" over passive sightseeing [40].

**Reimagining educational experiences as value-centered participation.** To address the finding that educational experience influences cultural identity primarily via emotional resonance, DMOs should shift from passive knowledge dissemination to active, value-focused engagement. Initiatives could include offering interactive workshops where tourists learn about and apply Mazu's core values through hands-on activities, and training interpretive guides to frame historical facts within frameworks of personal relevance (e.g., connecting narratives to universal themes of family and safety).

**Strengthening the nexus between cultural identity and place identity.** Since cultural identity is the direct antecedent to place identity, DMOs should explicitly reinforce the link between Mazu culture and its birthplace, Meizhou Island. This can be achieved by developing "root-seeking" programs that highlight Meizhou Island's irreplaceable role in Mazu's history (e.g., guided tours of the original Mazu temple, storytelling sessions about Mazu's birth) and by creating souvenirs or commemorative experiences that symbolize the culture-place bond (e.g., personalized "Mazu blessing certificates" inscribed with Meizhou Island's name, photo opportunities at iconic cultural landmarks). These strategies help transform Meizhou Island from a destination into a perceived symbolic home for Mazu culture, thereby strengthening place identity.

## Conclusion

The psychological pathway from experiential stimuli to identity formation within Mazu folk-religious tourism has been successfully mapped in this study. The empirical findings directly address the research objectives by confirming a sequential mediation mechanism: the four dimensions of 4E cultural tourism experiences (with escapist being most potent) significantly enhance tourists' emotional resonance; this profound affective state, in turn, is pivotal for strengthening cultural identity; ultimately, this strengthened cultural identity is the direct driver of place identity.

The primary theoretical contribution of this work lies in the empirical validation and specification of the "experience → emotion → cultural identity → place identity" chain. This delineation moves beyond establishing simple associations to clarify the sequential order and necessity of the mediating states. Consequently, a refined elaboration of the S-O-R framework for cultural tourism contexts is provided, clarifying that the internal "organism" process is best understood as a two-stage sequence where affective alignment precedes and facilitates identity internalization. This model offers an integrative lens for understanding how multidimensional tourist experiences are cognitively and affectively processed to forge deep-seated bonds with both a cultural system and its physical place.

In sum, a validated, context-specific model that bridges experiential design with profound psychological outcomes is presented. The findings establish that in folk-religious tourism, identity is not a direct result of experience but is built through a sequential psychological process where emotional resonance and cultural identity serve as indispensable, chained mediators. This understanding forms a critical foundation for both advancing theoretical discourse and informing sustainable heritage tourism management.

## Limitations and future research

This study's limitations chart a course for future inquiry. First, the use of a convenience sample from a single site suggests the need to test the model's generalizability across diverse folk-religious contexts (e.g., Shintoism, Hinduism) and tourist profiles (e.g., international visitors). Second, the cross-sectional design invites longitudinal research to trace the evolution

of these psychological constructs over time. Third, while our model focused on core psychological mechanisms and pre-liminary checks did not show strong demographic effects, future studies could explicitly examine demographic variables (e.g., age, income) as moderators to explore subgroup differences. Finally, a promising avenue is to investigate potential disruptors of the chain, such as "negative resonance" arising from perceived over-commercialization, which may moderate the relationship between experiences and identity outcomes.

## Supporting information

**S1 Appendix. Measurement item details for the study.** This table contains the complete wording of all survey items, along with their descriptive statistics (means and standard deviations), standardized factor loadings, and standard errors. (DOCX)

**S2 File. Research data.** Raw survey data from 583 tourists. (XLS)

## Author contributions

**Conceptualization:** Mian Wang, Hee Seung Lee, Wen Qiang Chen.

**Data curation:** Mian Wang, Hee Seung Lee, Wen Qiang Chen.

**Formal analysis:** Mian Wang, Hee Seung Lee, Wen Qiang Chen.

**Investigation:** Mian Wang, Wen Qiang Chen.

**Methodology:** Mian Wang, Hee Seung Lee, Wen Qiang Chen.

**Software:** Mian Wang, Wen Qiang Chen.

**Supervision:** Mian Wang, Hee Seung Lee, Wen Qiang Chen.

**Writing – original draft:** Mian Wang, Hee Seung Lee, Wen Qiang Chen.

**Writing – review & editing:** Mian Wang, Hee Seung Lee, Wen Qiang Chen.

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
