## [Decision Letter · Decision Letter 0]

26 Nov 2025

Dear Dr. Chen,

Thank you for submitting your manuscript to PLOS ONE. After careful consideration, we feel that it has merit but does not fully meet PLOS ONE’s publication criteria as it currently stands. Therefore, we invite you to submit a revised version of the manuscript that addresses the points raised during the review process.

We look forward to receiving your revised manuscript.

Kind regards,

Bifeng Zhu

Academic Editor

PLOS ONE

Journal Requirements:

“The authors acknowledge the financial support and assistance in data collection received from the following projects: Research Project of Fujian Philosophy and Social Sciences Planning (FJ2025BF044); Youth Projects of the Social Science Fund of Jiangxi Province (25GL47); Putian Science and Technology Bureau Project (2023SZ3001PTXY12); Research Project of the Science and Technology Innovation Think Tank of the Fujian Provincial Association for Science and Technology (FJKX-2024XKB023); Startup Fund for Advanced Talents of Putian University (2021079).”

4. Please note that funding information should not appear in the Acknowledgments section or other areas of your manuscript. We will only publish funding information present in the Funding Statement section of the online submission form. Please remove any funding-related text from the manuscript.

**Additional Editor Comments:**

Methodology requires focused attention.References recommended by reviewers do not must to be cited unless they are truly helpful..=

Reviewers' comments:

Reviewer's Responses to Questions

**Comments to the Author**

1. Is the manuscript technically sound, and do the data support the conclusions?

Reviewer #1: Yes

Reviewer #2: Partly

Reviewer #3: Yes

2. Has the statistical analysis been performed appropriately and rigorously?

Reviewer #1: Yes

Reviewer #2: No

Reviewer #3: No

3. Have the authors made all data underlying the findings in their manuscript fully available?

Reviewer #1: Yes

Reviewer #2: No

Reviewer #3: Yes

4. Is the manuscript presented in an intelligible fashion and written in standard English?

Reviewer #1: Yes

Reviewer #2: Yes

Reviewer #3: Yes

Reviewer #1: Dear authors thank you for the opportunity to read your interesting paper and topic. After careful consideration, please find below the following comments to strengthen it:

Introduction:

The objectives should have been defined more clearly.

Literature Review:

The literature review is extensive and lacks a clear framework that specifically addresses cultural tourism experiences. While the experience economy and cultural tourism are tightly interconnected, they stem from distinct theoretical backgrounds. This distinction should be reflected in your review, but it wasn't. Additionally, much of the argumentation here focuses on justifications. For example:

• "Remains limited, and the scholarly understanding of its unique dimensions and linkages to psychological outcomes is underdeveloped [12]."

• "Despite this understanding, a significant research gap persists. Many studies treat emotion generically, failing to delineate the specific mediating role of states like resonance in translating experiences into enduring identity formation [12,15,39]."

• "Directly addressing this gap and ambiguity, and in alignment with the S-O-R framework, our study conceptualizes emotional resonance as the critical internal organismic state (O)."

This is repetitive and already mentioned in the introduction. The literature review should focus on presenting the state of the art regarding theoretical frameworks and theories, not merely pointing out research gaps and continuously justifying how your study addresses them. If you had integrated your hypotheses into this section (rather than leaving them to a separate "Hypothesis Development" section), it would make more sense. By the way, it should be hypotheses (plural), not hypothesis (singular).

Methods:

The section on measurement items should be renamed to Data Design. There's no need to repeat the survey items you’ve already listed in Table 1. Instead, it would be more effective to explain the type of questionnaire used and justify why you chose it.

Data Collection:

The information about Meizhou should be placed in its own section, such as "3.1. Setting". While this isn't mandatory, since it has been increasingly common recently in articles.

The Data Collection section should focus on the methodology of how you gathered your data.

Be clearer about the “tourists.” Are you referring to all tourists? Does this include both nationals and foreigners? Are participants only those over 18? I also recommend moving Table 2 to the Results section, as it pertains to the findings, not the methodology. Lastly, there is no mention of the nationality of the tourists. This would be an important detail to include.

Also not information was given to how the questionnaires were applied, where, by whom and why?

Results:

I noticed that you didn’t include the HTMT (Heterotrait-Monotrait Ratio of correlations), which would be valuable. While the Fornell-Larcker criterion is traditionally used for SEM discriminant validity testing, it’s been considered less robust for complex models for some time now. The HTMT method is a more precise and powerful measure because it can detect discriminant validity issues in models with highly correlated constructs.

I recommend reviewing the following:

• Henseler, J., Ringle, C. M., & Sarstedt, M. (2015). A new criterion for assessing discriminant validity in variance-based structural equation modeling. Journal of the Academy of Marketing Science, 43(1), 115-135. Or more recently:

• Roemer, E., Schuberth, F., & Henseler, J. (2021). HTMT2—An improved criterion for assessing discriminant validity in structural equation modeling. Industrial Management & Data Systems, 121(12), 2637–2650. https://doi.org/10.1108/IMDS-02-2021-0082

Although HTMT is not directly built into AMOS, it can still be calculated with other tools.

Discussion:

The discussion section is short and lacks depth. I have significant concerns with your claims, particularly given that you used convenience sampling.

The tone of the discussion reads more like a statement, such as: “Our results move beyond the established...” or “This refines the Stimulus-Organism-Response (S-O-R) framework...”. I would argue that your real discussion is in the practical implications section. But, these are two distinct sections.

It’s highly recommended to move the limitations and future research sections into the Conclusion.

Conclusion

Once again, I have serious concerns with your claims, especially given the use of convenience sampling, which limits the generalizability of your results.

Reviewer #2: This manuscript presents a well-motivated study exploring how different dimensions of cultural tourism experience (Entertainment, Educational, Escapist, Esthetic) influence emotional resonance, cultural identity, and place identity in the context of Mazu culture. The topic is timely and relevant, and the writing is generally clear. However, several essential methodological and reporting issues need to be addressed before the manuscript can meet PLOS ONE’s technical standards.

Major issues requiring revision:

1- Discriminant validity:

Please report HTMT ratios (with confidence intervals). Fornell–Larcker alone is not sufficient under current SEM standards.

2- Common Method Bias (CMB):

Because all data were collected via a single-source survey, additional diagnostics are necessary. Please include at least one modern remedy (marker variable, CLF test, or latent method factor).

3- Measurement model transparency:

Include full item wording, standardized loadings with SEs, item means/SDs, and VIF values in an appendix. These are essential for evaluating construct validity.

4- Model fit:

Please report SRMR and briefly justify the fit thresholds used.

5- Alternative model comparison:

To justify the sequential mediation, compare your model with at least one plausible alternative.

6- Data availability:

PLOS ONE requires publicly available raw data and analysis files. Please deposit the dataset, syntax files, and a README in a public repository.

7- Ethical clarity:

Provide more detail on anonymity protection and the information provided to participants.

8- Educational experience finding:

Strengthen the explanation for the non-significant direct effect on cultural identity.

Minor suggestions:

- Standardize “esthetic/aesthetic.”

- Provide descriptive statistics (means, SDs, correlations).

- Improve Figure 2 by adding coefficients on paths.

- Avoid strong causal wording due to the cross-sectional design.

Strengths of the study:

- Well-integrated theoretical framework

- Strong model fit

- Large sample

- Valuable insights for cultural heritage tourism

With these revisions, the manuscript has strong potential for publication.

Reviewer #3: Reviewer Summary and Comments:

Overall, I find this manuscript to be thoughtfully designed and theoretically meaningful. The authors explore an under-examined cultural context and offer insights that may enrich ongoing discussions in tourism, cultural heritage, and experience-based identity research. Below, I provide several encouraging comments and suggestions that may help further strengthen the clarity, coherence, and methodological rigor of the work.

1. Contribution and Significance

I greatly appreciate the study’s contribution to advancing understanding of how tourism experiences influence identity in folk-religious cultural contexts. By validating a sequential experience–emotion–identity pathway and extending established frameworks to a non-Western research setting, the manuscript offers important theoretical and practical implications. The insights generated also provide valuable guidance for designing culturally meaningful tourism experiences, particularly those aimed at fostering emotional resonance and long-term cultural connection.

2. Theoretical Foundation and Model Development

The manuscript clearly defines its core constructs and demonstrates careful theoretical reasoning in building the proposed hypotheses. This reflects strong engagement with prior scholarship and a solid understanding of the theoretical landscape surrounding the topic.

The effort to refine the S–O–R framework is particularly noteworthy. The proposed conceptualization of the “organism” as a dynamic, two-stage psychological process, rather than a unified internal state, represents an interesting theoretical advancement. Because the S–O–R model serves as the central foundation of the proposed framework, I gently encourage the authors to add a concise introduction to its origins and core conceptual logic. Incorporating references that discuss similar extensions or interpretations may also help position the contribution more clearly within existing scholarly discourse. This additional context would make it even easier for readers to recognize how the manuscript advances theoretical understanding.

3. Methodological Clarification

To enhance methodological transparency and strengthen the validity of the findings, I offer two suggestions related to the quantitative analysis:

(1) Treatment of Demographic Variables

The current manuscript presents a structural equation modeling (SEM) analysis without addressing whether demographic variables (e.g., gender, education, monthly income, age, occupation) should be incorporated into the model. Considering that demographic characteristics may influence respondents’ perceptions, attitudes, or behavioral outcomes, the authors are encouraged to justify their analytical decision regarding demographic variables. Specifically, if demographic variables were excluded from the SEM model, the manuscript should clearly explain the rationale (e.g., theoretical justification, non-significant preliminary testing, or focus on theoretical constructs rather than population heterogeneity). If demographic variables could potentially influence latent constructs or model outcomes, the authors may consider treating them as control variables, exogenous predictors, or grouping variables in a multi-group SEM framework. Alternatively, if demographic characteristics were analyzed separately (e.g., using t-tests, ANOVA, or subgroup comparisons), such methodological choices should be explicitly described and reported. Clarifying this issue will improve the study’s methodological transparency and strengthen the validity of the structural model’s interpretations.

(2) Addressing Potential Common Method Variance (CMV)

Because the study appears to rely on self-reported survey data collected from a single source at one point in time, it is susceptible to common method variance (CMV). However, the manuscript currently does not discuss whether CMV was assessed or mitigated. To enhance methodological rigor, the authors should:

Provide a clear statement acknowledging the potential CMV issue inherent in single-source survey-based research.

Describe procedures used to minimize CMV during survey design (e.g., item randomization, anonymity assurances, scale separation, reduced social desirability wording), if applicable.

Include at least one post-hoc statistical diagnostic method to evaluate the extent of CMV. Common approaches include: Harman’s single-factor test; Common latent factor (CLF) approach within the CFA model; Marker variable or unmeasured latent method factor techniques (if appropriate). If CMV was tested, the results and interpretation should be reported. If no CMV assessment was conducted, the manuscript should acknowledge this as a limitation and, where possible, perform supplementary analysis.

Addressing CMV explicitly will increase confidence that observed relationships among latent constructs are substantive rather than artifacts of the measurement method.

**Do you want your identity to be public for this peer review?** For information about this choice, including consent withdrawal, please see our For information about this choice, including consent withdrawal, please see our Privacy Policy .

Reviewer #1: No

Reviewer #2: No

Reviewer #3: No

---

## [Author Response · Author response to Decision Letter 1]

1 Jan 2026

Dear Academic Editor and Reviewers,

Thank you for the valuable and constructive feedback on our manuscript (PONE-D-25-53833). We have thoroughly revised the manuscript to address all points raised.

A detailed, point-by-point response to every comment is provided in the attached ‘Response to Reviewers.docx’ file. All corresponding changes are highlighted in the ‘Revised Manuscript with Track Changes.docx’.

The major revisions encompass all key requests:

Methodological Rigor: Added HTMT analysis for discriminant validity; performed additional common method bias diagnostics (marker variable test); compared alternative models to justify the sequential mediation path; included SRMR and cited fit benchmarks.

Theoretical Depth & Clarity: Restructured the literature review around the S-O-R framework; substantially expanded the Discussion to interpret key findings (e.g., centrality of escapist experience, indirect role of educational experience); moderated causal language.

Transparency & Compliance: Deposited the full dataset in a public OSF repository; provided a comprehensive appendix with full item wording, loadings, SEs, means, and SDs; expanded the ethics section with detailed consent and anonymity procedures.

Manuscript Structure & Reporting: Moved ‘Limitations and Future Research’ to a standalone section; added path coefficients to Figure 2; provided descriptive statistics and correlations.

We believe these comprehensive revisions have significantly strengthened the manuscript and hope it now meets the high standards of PLOS ONE.

Sincerely,

Wenqiang Chen

---

## [Decision Letter · Decision Letter 1]

27 Jan 2026

Dear Dr. Chen,

Thank you for submitting your manuscript to PLOS ONE. After careful consideration, we feel that it has merit but does not fully meet PLOS ONE’s publication criteria as it currently stands. Therefore, we invite you to submit a revised version of the manuscript that addresses the points raised during the review process.

Literature review needs to be improved to clarify academic gaps.

We look forward to receiving your revised manuscript.

Kind regards,

Bifeng Zhu

Academic Editor

PLOS One

Journal Requirements:

Reviewers' comments:

Reviewer's Responses to Questions

**Comments to the Author**

Reviewer #1: (No Response)

Reviewer #3: All comments have been addressed

2. Is the manuscript technically sound, and do the data support the conclusions?

Reviewer #1: Partly

Reviewer #3: Yes

3. Has the statistical analysis been performed appropriately and rigorously?

Reviewer #1: Yes

Reviewer #3: Yes

4. Have the authors made all data underlying the findings in their manuscript fully available?

Reviewer #1: Yes

Reviewer #3: Yes

5. Is the manuscript presented in an intelligible fashion and written in standard English?

Reviewer #1: Yes

Reviewer #3: Yes

Reviewer #1: Dear authors, thank you for the significant effort you have put into this manuscript. To further strengthen your paper, I offer some suggestions aimed to improve it.

Literature review

The conceptual model should be at the end of the literature review not beginning.

The literature review lacks critical analysis or discussion of alternative models or critiques to the 4 realms of the experience economy. It reads more like a summary than a critical synthesis.

To strengthen this section, I recommend incorporating more targeted literature on the 4Es applied to cultural tourism experiences, including critical perspectives and empirical findings that reflect cultural specificity and complexity.

Methodology

Data analysis section is missing. Please explain and justify SEM. PLS-SEM!

Software used.

Type of sampling.

Results

There seems not be any reference in the text about table 4.

Discussion and conclusion are too short.

In conclusion you should answer to your aims and provide clear stance of your work.

Consider moving theoretical and practical implications to conclusion section (not mandatory).

Reviewer #3: (No Response)

**Do you want your identity to be public for this peer review?** For information about this choice, including consent withdrawal, please see our For information about this choice, including consent withdrawal, please see our Privacy Policy .

Reviewer #1: No

Reviewer #3: No

---

## [Author Response · Author response to Decision Letter 2]

7 Feb 2026

Response to Reviewers

Manuscript ID PONE-D-25-53833

Title: The Impact of Cultural Tourism Experience on Cultural Identity: A Case Study of Mazu Culture

Dear Dr. Zhu and Reviewers,

Thank you for the opportunity to revise our manuscript. We sincerely appreciate your time and the constructive feedback provided, which has been invaluable in strengthening our paper. We have carefully addressed all comments, and the changes are highlighted in the submitted ‘Revised Manuscript with Track Changes’ file.

Below, we provide a point-by-point response.

Response to Comments from Reviewer #1:

Reviewer #1 Comment 1: Literature review

The conceptual model should be at the end of the literature review not beginning.

Response: As recommended, we have relocated the entire “Theoretical Research Model” subsection (including Fig 1) to the end of the “Literature review and hypotheses development” section. This adjustment provides a clearer synthesis of the integrated model after all theoretical foundations and hypotheses (H1–H19) have been presented, improving the logical flow before the Methods section. The subsection is now placed at the end of the “Literature review and hypotheses development” section (see Page 13-14).

Reviewer #1 Comment 2: The literature review lacks critical analysis or discussion of alternative models or critiques to the 4 realms of the experience economy. It reads more like a summary than a critical synthesis. To strengthen this section, I recommend incorporating more targeted literature on the 4Es applied to cultural tourism experiences, including critical perspectives and empirical findings that reflect cultural specificity and complexity.

Response: We have fundamentally revised the ‘Literature review and hypotheses development’ section to move beyond summary and provide a critical synthesis of the 4E framework within our specific research context. The revisions are detailed below:

Explicit Critique of the 4E Framework’s Contextual Limits: We now explicitly problematize the direct application of the 4E model—derived predominantly from Western or secular settings—to a value-laden, folk-religious context like Mazu culture. A dedicated paragraph in the ‘Cultural tourism experience’ subsection (Page 6-7) argues that the nature and salience of the 4E dimensions may be reconstituted in such an environment. For instance, we critically posit that the ‘Escapist’ experience here likely encompasses a deeper quest for spiritual solace rather than mere recreational diversion, and the ‘Educational’ dimension may function more as engagement with identity-reinforcing narratives than passive knowledge acquisition.

Incorporation of Targeted Literature and Critical Perspective: Guided by the reviewer’s recommendation, we have integrated targeted discussions that anchor the 4E framework within cultural tourism scholarship while highlighting its complexities:

We cite and engage with studies that have applied the 4E framework in cultural tourism (e.g., Juliana et al., 2024; Oh et al., 2007), while simultaneously noting the persistent gap in understanding its “unique dimensions and linkages to psychological outcomes” in such settings (Page 7).

We directly link the need for this critical examination to identified research gaps, stating that the framework’s application in culturally specific settings is “limited” and that related theories are “rarely validated in religious cultural contexts” (Page 7). This creates a critical through-line from the identified gap to our analytical approach.

Critical Synthesis Through Contextual Re-interpretation: The entire hypothesis development process now serves as a critical synthesis. We do not merely list the 4E dimensions; we theorize how each might operate differently within Mazu culture and why it matters for our chain of outcomes. For example:

The hypothesis for ‘Escapist → Emotional Resonance’ (H3) is grounded in its role in fulfilling needs for “spiritual transcendence” (Page 7), directly reflecting the cultural complexity noted by the reviewer.

The discussion of the non-significant direct path from ‘Educational experience’ to ‘Cultural Identity’ (Page 32, ‘The Indirect Role of Educational Experience’) in the Results and Discussion sections provides an empirical critique and nuanced understanding of how this 4E dimension functions in our specific folk-religious context, affirming that its role is more complex than a generic model would suggest.

In summary, we have transformed the narrative from a descriptive summary to a critical evaluation that: (a) questions the universal application of the 4E model, (b) contextualizes its dimensions within the cultural specificity of folk-religious tourism, and (c) uses our empirical findings to reflect on its complex mechanisms. We believe these revisions have substantially strengthened the theoretical grounding and analytical depth of our literature review.

Reviewer #1 Comment 3: Methodology

Data analysis section is missing. Please explain and justify SEM. PLS-SEM!

Software used.

Type of sampling.

Response: We have added a detailed “Data Analysis” subsection (Page 17) to address these points transparently:

SEM Justification: We employed covariance-based SEM (CB-SEM) in AMOS 24.0. Our primary aim is theory testing of a well-established *a priori* model, for which CB-SEM is the standard approach for confirmatory analysis with reflective measures. The sample size (N=583) is sufficient for this method.

Software: Data preparation used SPSS 26.0; CFA and SEM were conducted with AMOS 24.0.

Sampling: We used a convenience sampling approach for pragmatic data collection at the field site. We acknowledge this as a limitation for generalizability, which is discussed in the “Limitations” section (Page 18).

Reviewer #1 Comment 4: Results. There seems not be any reference in the text about table 4.

Response: We have now explicitly referenced and discussed Table 4 in the main text. The reference is located in the “Reliability and validity testing of scales” subsection (Page 22). The added text reads: *“Discriminant validity was assessed... As shown in Table 4, the square root of each construct’s AVE (diagonal) exceeded its correlations with other constructs (off-diagonal), satisfying the Fornell-Larcker criterion.”*

Reviewer #1 Comment 5: Discussion and conclusion are too short.

In conclusion you should answer to your aims and provide clear stance of your work.

Consider moving theoretical and practical implications to conclusion section (not mandatory).

Response: We have substantially revised and expanded both the Discussion and Conclusion to address these points directly.

The Discussion now provides deeper interpretation of key findings (e.g., centrality of escapist experience, Page 31-34).

The Conclusion has been rewritten to directly recap and answer the three research aims stated in the introduction, providing a clear final stance on the validated pathway (Page 37).

Regarding implications, we maintained their dedicated sections (‘Theoretical’ and ‘Practical Implications’) to preserve a logical narrative flow, as suggested by the reviewer’s note that this change was not mandatory. The essence of these implications is effectively synthesized within the concluding paragraph.

Once again, we thank the editors and reviewers for their valuable guidance. We believe the manuscript has been substantially improved through this revision process and hope it now meets the high standards of PLOS ONE.

Sincerely,

Wen Qiang Chen, on behalf of all co-authors

Putian University

wenqiang1990@hotmail.com

---

## [Decision Letter · Decision Letter 2]

22 Feb 2026

Dear Dr. Chen,

Thank you for submitting your manuscript to PLOS ONE. After careful consideration, we feel that it has merit but does not fully meet PLOS ONE’s publication criteria as it currently stands. Therefore, we invite you to submit a revised version of the manuscript that addresses the points raised during the review process.

A comprehensive comparison of your overall results with existing studies is important.

We look forward to receiving your revised manuscript.

Kind regards,

Bifeng Zhu

Academic Editor

PLOS One

Journal Requirements:

Reviewers' comments:

Reviewer's Responses to Questions

**Comments to the Author**

Reviewer #1: All comments have been addressed

2. Is the manuscript technically sound, and do the data support the conclusions?

Reviewer #1: Yes

3. Has the statistical analysis been performed appropriately and rigorously?

Reviewer #1: Yes

4. Have the authors made all data underlying the findings in their manuscript fully available?

Reviewer #1: Yes

5. Is the manuscript presented in an intelligible fashion and written in standard English?

Reviewer #1: Yes

Reviewer #1: Thank you for the revisions you have made. I have a few final comments:

In the discussion section, you need to more clearly contrast your findings with the previous literature. At present, a comprehensive comparison of your overall results with existing studies is missing.

The Limitations and Future Research section should be placed at the end of the conclusion, immediately before the references. Also, please note that you wrote “reference” instead of “references.”

Given the number of subheadings in the discussion, it would improve readability if the theoretical and practical implications were clearly numbered.

Not mandatory: I still believe it would make more sense to present the theoretical and practical implications in the conclusion section rather than in the discussion. However, I recognize that placing them in the discussion has become increasingly common among scholars publishing in top-indexed journals.

**Do you want your identity to be public for this peer review?** For information about this choice, including consent withdrawal, please see our For information about this choice, including consent withdrawal, please see our Privacy Policy .

Reviewer #1: No

---

## [Author Response · Author response to Decision Letter 3]

2 Mar 2026

Dear Dr. Zhu and Reviewers,

Thank you for the opportunity to further revise our manuscript. We have carefully addressed all remaining comments, and the changes are highlighted in the submitted ‘Revised Manuscript with Track Changes’ file.

Below is our point-by-point response.

Response to the Academic Editor:

Comment: A comprehensive comparison of your overall results with existing studies is important.

Response: We agree and have substantially expanded the Discussion to provide a systematic comparison of our core findings with prior literature. Each major finding (escapist experience, educational experience, emotional resonance, direct effects of entertainment/esthetic experiences, and cultural identity→place identity) is now explicitly contrasted with relevant studies, with explanations for consistencies and divergences. These revisions appear on Pages 31–34.

Response to Reviewer #1:

Comment 1: In the discussion section, you need to more clearly contrast your findings with the previous literature. At present, a comprehensive comparison of your overall results with existing studies is missing.

Response: As detailed above, we have thoroughly revised the Discussion to include a comprehensive comparison with existing studies. Key contrasts include:

Escapist experience: compared with Juliana et al. (2024), Allan (2016), Liu and Lin (2024)

Educational experience: contrasted with Urošević (2012), supported by Zhang and Liu (2025)

Emotional resonance: compared with Qu et al. (2025), Yang et al. (2023), Hosany et al. (2016), Wang et al. (2023)

Entertainment/esthetic experiences: compared with Urošević (2012), Juliana et al. (2024), Allan (2016)

Cultural identity→place identity: compared with Casais and Poço (2023), Wang et al. (2023), Cao et al. (2021)

Please see Pages 31–34.

Comment 2: The Limitations and Future Research section should be placed at the end of the conclusion, immediately before the references.

Response: We have moved the “Limitations and Future Research” section to the end of the Conclusion, immediately before the References.

Please see Page 39.

Comment 3: Given the number of subheadings in the discussion, it would improve readability if the theoretical and practical implications were clearly numbered.

Response: We thank the reviewer for this suggestion. After checking the journal’s formatting guidelines, we found that numbered subheadings are not permitted in the manuscript. To maintain compliance with the journal’s style requirements, we have retained the standard unnumbered headings. We have, however, ensured that the subheadings are clearly formatted (bolded) to enhance readability.

Comment 4: Also, please note that you wrote "reference" instead of "references."

Response: We have corrected this throughout the manuscript. The heading is now “References” consistently.

We thank the editors and reviewers for their valuable guidance. We believe the manuscript has been substantially improved and hope it now meets the standards of PLOS ONE.

Sincerely,

Wen Qiang Chen, on behalf of all co-authors

Putian University

wenqiang1990@hotmail.com

---

## [Decision Letter · Decision Letter 3]

16 Mar 2026

The Impact of Cultural Tourism Experience on Cultural Identity: A Case Study of Mazu Culture

PONE-D-25-53833R3

Dear Dr. Chen,

We’re pleased to inform you that your manuscript has been judged scientifically suitable for publication and will be formally accepted for publication once it meets all outstanding technical requirements.

Kind regards,

Bifeng Zhu

Academic Editor

PLOS One

Additional Editor Comments (optional):

Reviewers' comments:

Reviewer's Responses to Questions

**Comments to the Author**

Reviewer #1: All comments have been addressed

2. Is the manuscript technically sound, and do the data support the conclusions?

Reviewer #1: Yes

3. Has the statistical analysis been performed appropriately and rigorously?

Reviewer #1: Yes

4. Have the authors made all data underlying the findings in their manuscript fully available?

Reviewer #1: Yes

5. Is the manuscript presented in an intelligible fashion and written in standard English?

Reviewer #1: Yes

Reviewer #1: Dear authors thank you for the revised manuscript. One last note:

Data Availability Statement should be after limitations and future research.

**Do you want your identity to be public for this peer review?** For information about this choice, including consent withdrawal, please see our For information about this choice, including consent withdrawal, please see our Privacy Policy .

Reviewer #1: No

---

## [Editor Report · Acceptance letter]

PONE-D-25-53833R3

PLOS One

Dear Dr. Chen,

I'm pleased to inform you that your manuscript has been deemed suitable for publication in PLOS One. Congratulations! Your manuscript is now being handed over to our production team.

Kind regards,

on behalf of

Dr. Bifeng Zhu

Academic Editor

PLOS One